# Reliable and Trustworthy Machine Learning for Health Using Dataset Shift Detection

**Chunjong Park, Anas Awadalla, Tadayoshi Kohno, Shwetak Patel**
Paul G. Allen School of Computer Science & Engineering
University of Washington
{cjparkuw, anasa2, yoshi, shwetak}@cs.washington.edu

## Abstract

Unpredictable ML model behavior on unseen data, especially in the health domain, raises serious concerns about its safety as repercussions for mistakes can be fatal. In this paper, we explore the feasibility of using state-of-the-art out-of-distribution detectors for reliable and trustworthy diagnostic predictions. We select publicly available deep learning models relating to various health conditions (e.g., skin cancer, lung sound, and Parkinson's disease) using various input data types (e.g., image, audio, and motion data). We demonstrate that these models show unreasonable predictions on out-of-distribution datasets. We show that Mahalanobis distance- and Gram matrices-based out-of-distribution detection methods are able to detect out-of-distribution data with high accuracy for the health models that operate on different modalities. We then translate the out-of-distribution score into a human interpretable CONFIDENCE SCORE to investigate its effect on the users' interaction with health ML applications. Our user study shows that the CONFIDENCE SCORE helped the participants only trust the results with a high score to make a medical decision and disregard results with a low score. Through this work, we demonstrate that dataset shift is a critical piece of information for high-stake ML applications, such as medical diagnosis and healthcare, to provide reliable and trustworthy predictions to the users.

## 1 Introduction

Advances in artificial intelligence and machine learning have made medical diagnostic and screening tools more accurate and accessible. AI-powered diagnostic tools [5, 14] are intended to assist medical personnel by making unbiased decision based on thousands of examples. In recent years, these models [15, 44, 48, 32] are even becoming available to consumers through the growth of mobile health with the intention of expediting diagnoses through increasingly frequent testing. Moreover, mobile health [4, 50] aims to improve access to medical expertise for those who are uninsured or live far away from hospitals.

Despite the potential benefits of health AI systems, there are concerns about their performance in real-world settings. Data-driven models learn from examples, making them heavily reliant on the data upon which they have been trained. However, datasets often fail to get complete coverage over a domain, particularly for emerging datasets; when new pulmonary diseases (e.g., MERS and COVID-19) emerge, a pulmonary classifier trained on the existing lung sounds would not be able to interpret sound of the new diseases. Previous work [42, 78] has found that machine learning models behave unpredictably on the unseen data. This problem [4, 69] is especially critical for medical diagnostic and screening tools since there are significant repercussions for mistakes.

Researchers have proposed methods to estimate the uncertainty of a machine learning models' predictions based on the input [29, 41, 40, 64, 43]. Out-of-distribution detection methods can distinguish whether the input lies within the distribution of the training dataset, with out-of-distribution

35th Conference on Neural Information Processing Systems (NeurIPS 2021).

data leading to less reliable prediction results. However, such important information has not been widely explored in the context of health applications. When health applications are put into the hands of consumers with limited understanding of the underlying algorithms, they may upload poor quality data that lies outside the distribution of the data that was collected by experts. For example, consumers who are using a health application that involves image processing may take photographs in poor lighting conditions or framing of the target object. Even when the data is high-quality, it may be captured with a smartphone that has different hardware specifications than the devices that were used to collect the training dataset. Unless the models are explicitly designed or trained to detect invalid data, the models will incorrectly produce a diagnostically meaningless result.

In this work, we explore the utility of out-of-distribution detection for improving model performance and user-perceived trustworthiness of health-related models. We first benchmark our approach using publicly available deep learning models relating to various medical challenges and sensing domains — images for skin lesion classification, motion data for Parkinson's disease severity, and audio for lung sound classification. After demonstrating that these models are susceptible to dataset shift, we demonstrate that the state-of-the-art out-of-distribution detectors can effectively exclude such data with over 95% detection accuracy in most cases. We then explore the implications of this detector on user-perceived trustworthiness of the health models. After translating the out-of-distribution score into a human-interpretable metric, CONFIDENCE SCORE, we found that showing this information to end-users improved the user-perceived trustworthiness of the models. Furthermore, participants stated that they were more willing to make medical decisions based on models when they were shown the certainty metric. Our contributions in this work are as follows:

- We identify and quantify the limitations of current health deep learning models when encountered with unseen data,
- We evaluate the utility of out-of-distribution detection on various data types (e.g., image, audio, motion) for medical screening and diagnosis, and
- We evaluate the impact that dataset shift information has on user-perceived trustworthiness of health diagnostic results.

## 2 Related Work

**Machine Learning-based Health Screening and Diagnosis** In recent years, machine learning has been widely used for medical diagnosis and screening tool to help doctors diagnosis patients easier, faster, and more accurately. Machine learning models that learn from large-scale medical datasets are able to detect various symptoms and conditions, including mental health [26, 68], retinal disease [14], lung cancer [5]. With the increasing ubiquity of smartphone and advances in its computing power, machine learning-based health screening can be done on mobile devices. Various machine learning-based mobile health applications have been proposed to detect health conditions (e.g., traumatic brain injury [49], pancreatic cancer [48], jaundice [15]) and vital signals (e.g., heart rate [44], respiratory rate [44], heart rate variability [32], blood pressure [65], SpO2 [31]). Such mobile health applications can benefit nurses, health workers, and the general population for easier medical screening.

While the health machine learning models show high accuracy on their own test datasets, their performance is questionable in real-world settings where the input data can vary drastically, resulting in unreliable prediction results [70, 63]. Researchers have investigated the dataset shift problem for medical imaging (e.g., x-ray [11, 10], fundus eye images [11], CT scans [74], dermatology [63, 56]), focusing on developing and evaluating out-of-distribution detection methods for specific domains. However, as more consumer-facing health applications are available in the market, this issue can lead the users to make medical decision based on incorrect results. In this work, we aim to explore ways to leverage dataset shift information to make the health machine learning models more reliable and trustworthy to the users.

**Dataset Shift Detection** Recently researchers have proposed various methods to estimate the models' uncertainty due to dataset shift. The proposed methods leverage the output of the models to effectively detect *out-of-distribution* input that are different from the known distribution, *in-distribution*. Softmax confidence [29] has been the baseline for the out-of-distribution detection. Several work has been proposed for out-of-distribution detection using deep ensemble [38], Mahalanobis distance [40], Gram matrices [64], energy score [43], temperature scaling [41, 64], input perturbation [41, 40], mean and variance of channels activations [58]. Alternate training strategies [30, 39, 47, 52] have

been proposed to enable model to detect out-of-distribution. Generative models [54, 60, 53, 77] are proposed to detect out-of-distribution examples. However, many approaches require re-training and re-designing of the models and prior knowledge of out-of-distribution datasets; it is not realistic to apply these methods to the existing models. In this work, we explore Mahalanobis distance- [40], Gram matrices- [64], and energy-based [43] out-of-distribution detection methods for reliable and trustworthy machine learning for health since these methods show reasonable out-of-distribution detection performance, do not require retraining or prior knowledge of out-of-distribution datasets, and work on pre-trained discrimitive classifiers.

**Trustworthy AI** Machine learning systems are deployed in real-world settings to billions of users, making significant impacts on high-stake decision making such as healthcare, policy, economy, and transportation. Failures in machine learning systems can cause fatal consequences and building trustworthy AI is one of the most important problems in machine learning community. In recent years, there are active and ongoing efforts aimed at making machine learning systems causal [6, 55], explainable [2, 36, 45, 34, 35], fair [3, 59, 17], robust [18, 21, 28, 19, 72], and privacy-preserving [1, 57, 51, 71]. This work contributes to trustworthy AI by improving reliability and user-perceived trustworthiness of machine learning for health using estimated uncertainty. Bhatt et al. [8] proposed to leverage uncertainty for users making decision and placing trust in machine learning models. This work explores similar approach where we adopt out-of-distribution detection as a method to measure uncertainty. We took a step further to investigate and quantify its effect in improving reliability and trustworthiness in the context of machine learning for health.

## 3 Background: Dataset Shift Detection Methods

We aim to leverage state-of-the-art out-of-distribution detection methods [40, 64, 43] in the health domain for users to safely use health deep learning models. We selected three out-of-distribution methods that show high accuracy on different out-of-distribution datasets, do not require re-training and prior knowledge of out-of-distribution datasets, and work on pre-trained discrimitive classifiers. These characteristics are important to help developers or other stakeholders (e.g., regulators, auditors, platforms) easily adopt out-of-distribution detectors to any pre-trained models. In this section, we provide background on each out-of-distribution detection methods.

### 3.1 Mahalanobis Distance-Based Out-of-Distribution Detection

Mahalanobis distance is used to measure the proximity of a point to a certain Gaussian distribution. In the Mahalanobis distance-based out-of-distribution detector [40], this property is used to represent each class's samples at each layer of a network as a class conditional Gaussian distribution with mean $\hat{\mu}_{cl}$ and co-variance $\hat{\Sigma}_{cl}$, where $c$ indicates the class and $l$ indicates the layer in the model. Given a sample input $x$ to a neural network, for each layer, it computes the minimum layer-wise class conditional Mahalanobis distances for $x$. That is, for each layer, it finds the Mahalanobis distance associated with the closest class to $x$. In other words, this is equivalent to $M(x) = \max_c - (f(x) - \mu_c)^T \Sigma^{-1} (f(x) - \mu_c)$. The authors have demonstrated that adding small noise to the input can help better distinguish between in-distribution and out-of-distribution data. As the authors suggested, for the real-world setting where the out-of-distribution datasets are generally not available, we obtain the input noise magnitude by generating adversarial samples generated by FGSM [27].

### 3.2 Gram Matrices-Based Out-of-Distribution Detection

Gram matrices are used to compute pairwise feature correlations and encode stylistic attributes. For out-of-distribution detection [64], higher order Gram matrices are used to compute class-conditional bounds of feature correlations at all hidden layers of the network as higher order shows better out-of-distribution detection performance. Higher order Gram matrices is expressed as $G_l^P = (F_l^P F_l^{P^T})^{\frac{1}{P}}$, where $F_l$ is feature map at $l$-th layer and $P$ is order. All elements of Gram matrices of an input at each layer are compared against the prepossessed minimum and maximum Gram matrices element values from in-distribution dataset to obtain deviation. If the input data is predicted as a certain class, the minimum and maximum values of the corresponding class will be used for comparison. The comparison is done for each layer to obtain layerwise deviations. Then, the deviations are used to get a total deviation, which is defined by the normalized sum of layerwise deviations. Whether the input data is from out-of-distribution is determined with a threshold which is defined as 95% percentile of the total deviations of in-distribution energy score distribution.

## 3.3 Energy-Based Out-of-Distribution Detection

The energy-based out-of-distribution detector [43] seeks to provide an alternative scoring function to the softmax function that is less susceptible to over-confidence and therefore can better distinguish between in and out-of-distribution inputs. It takes a discriminative classifier $f(c)$ that maps input $x \in \mathbb{R}^D$ to logits, which are traditionally used to derive a categorical confidence score using a softmax function. It defines the energy function on the classifier as $E(x; f) = -T \cdot log \sum_i^K e^{f_i(x)/T}$, where $K$ is the number of classes in the model's output space and $T$ is a temperature parameter that can be used to alter the shape of the energy score distribution. Energy score threshold that distinguishes between in- and out-of-distribution samples is defined at the 95% percentile of the in-distribution samples.

## 4 Performance Evaluation

In this section, we demonstrate the performance degradation of the existing health models when encountered with out-of-distribution datasets highlighting that the existing models are vulnerable to dataset shift. We then evaluate the performance of state-of-the-art out-of-distribution detectors for distinguishing between in- and out-of-distribution examples. We have selected out-of-distribution datasets that consist of both near- and far-from-distribution samples that represent realistic use cases in the real-world settings. For mobile health applications that use mobile sensors for health screening, non-expert users are expected to input data collected by themselves. Unlike clinicians, who may either receive training on how to operate these mobile apps or may already understand what must be done to generate high-quality, non-expert consumers may collect relevant but low-quality data due to environmental factors or totally irrelevant data by mistake or a lack of understanding. To reflect these scenarios, we include out-of-distribution datasets caused by covariate shift, label shift, and open-set recognition. The covariate and label shifts aim to evaluate a model's performance when tested on data pertaining to the same classification task but from different data sources and environment. Open-set recognition evaluates the model's performance on new classes not included in the training set. In Table 2, we indicate dataset shift type for each out-of-distribution dataset.

### 4.1 Models and Datasets

**Skin lesion** A DenseNet-121 based skin lesion classifier [56] was used in this work. The model aims to classify an image into seven different skin lesions: actinic keratoses, basal cell carcinoma, benign keratosis, dermatofibroma, melanoma, melanocytic nevi and vascular lesions. The following datasets are used for training and evaluation:

- (**In-distribution**) HAM10000 [73, 12]: (CC BY-NC 4.0) A dataset containing 10,000 samples of dermatoscopic skin tumor images taken using different devices and from different populations. These tumors are part of 7 classes: actinic keratoses, basal cell carcinoma, benign keratosis-like lesions, dermatofibroma, melanoma, melanocytic nevi, and vascular lesions.

- (**Out-of-distribution**) ISIC 2017 [13]: (CC BY-NC 4.0) A previous version of the HAM100000 dataset which contains 2000 dermatoscopic skin tumor images labelled for binary classification. A tumor is labelled malignant if it corresponds to melanoma to benign if it corresponds to nevus or seborrheic keratosis.

- (**Out-of-distribution**) Face [16]: (CC BY 4.0) A dataset containing frontal view face images of 102 adults without making a neutral facial expression. Face images are personally identifiable information. But, all individuals gave signed consent for their images to be "used in lab-based and web-based studies in their original or altered forms and to illustrate research (e.g., in scientific journals, news media or presentations)."

- (**Out-of-distribution**) CIFAR-10 [37]: (MIT License) A common image classification benchmark with 10 non-medical classes (airplane, car, cat, dog, horse, bird, deer, ship, frog, truck) which contains 6,000 images per class.

**Lung Sound** A lung sound classification model [23] classifies normal lung sound, wheeze, and crackle from an audio sample. This model is based on ResNet-34 and uses spectograms of audio samples as inputs and outputs 4 lung sound classes (normal, wheezing, crackle, and wheezing + crackle).

- (**In-distribution**) ICBHI 2017 Respiratory Challenge [61]: A dataset collected using multiple microphones and stethoscopes containing 6898 samples normal lung sound, wheeze, and crackle audio

- (**Out-of-distribution**) Stethoscope [22]: (CC BY 4.0) A dataset containing stethoscope respitory sounds with 336 samples of normal breathing, wheeze, and crackle audio sounds. The dataset was collected using a 3M Littmann Electronic Stethoscope.

- (**Out-of-distribution**) AudioSet [24]: (CC BY 4.0) A large dataset of millions of sound labelled YouTube audio of which a portion of the dataset contains breathing, cough, and wheezing samples which we use to create a suitable out-of-distribution dataset for this model.

**Parkinson's Disease** This is a binary classification model [76] that showed highest performance in Parkinson's disease digital biomarker DREAM challenge [66]. The model uses accelerometer signals to detect tremors in a person's movement and outputs whether a participant has Parkinson's. This model consists of 5 1D-convolutional layers and a single output.

- (**In-distribution**) mPower [9]: (CC BY 4.0) A dataset contains 30-second accelerometer readings from 3,100 participants at rest for both healthy and Parkinson's patients. The dataset was used in Parkinson's disease digital biomarker DREAM challenge [66].

- (**Out-of-distribution**) Kaggle Parkinson's disease [25]: (CC0 1.0) A dataset with accelerometer readings from healthy participants simulate movements of Parkinson's patients.

- (**Out-of-distribution**) MotionSense [46]: (MIT License) A dataset contains accelerometer readings from 24 participants performing various activities (e.g., walking, jogging, sitting, standing, etc).

- (**Out-of-distribution**) MHEALTH [7]: An activity classification dataset which contains accelerometer readings from 10 participants executing various activities (e.g., standing, sitting, walking, cycling, etc).

## 4.2 Performance Impact by Dataset Shift

In evaluating the model's performance on the out-of-distribution dataset, we used pre-trained models from the previous work[1] [23] when the authors make it available. Otherwise, we trained the model in the same way specified in their work[2] [56, 76]. We trained skin lesion model [56] for 150 epochs using Adam optimizer with a learning rate of 0.0001 and weight decay of 0.2. For Parkinson's disease model [76], we trained for 50 epochs using Adam optimizer with a learning rate of 0.0005. The pre-trained lung sound model [23] is trained for 200 epochs using SGD optimizer with a learning rate of 0.001 and momentum of 0.9. For all of these models, we used an 80/20 split and applied the same preprocessing for train and test sets. All training and testing is done in a server (Intel Xeon 2.1GHz, 64GB, GeForce RTX 2080 Ti) from an internal cluster. We then ran inference on each dataset and calculated the classification accuracy for the datasets that have corresponding labels. For the datasets that do not have the same labels from the in-distribution, the accuracy could not be computed. Table 1 summarizes the classification accuracy for the models on in- and out-of-distribution datasets.

We generally observed a significant performance drop for all health machine learning models that are tested with out-of-distribution datasets. The models output unreasonable and arbitrary predictions on datasets that are not related health conditions. For example, skin lesion classifier predicts all face images as vascular lesions and CIFAR10 images as various types skin lesions. Similarly, Parkinson's disease classifier predicts significant portion of physical activities by health participants as tremor caused Parkinson's disease. For lung sound classification, ordinary sound events (e.g., speech, walking, laughing) are classified as a certain type of lung sounds (e.g., crackle, wheezing). When the models are evaluated on out-of-distribution datasets that have similar data characteristics to the in-distribution data(i.e., near-distribution datasets), all health models exhibit a performance decrease that ranges from 18% to 76%. This implies that the models are also sensitive to small dataset shift, such as datasets collected with different devices and in different environments. All of these failure scenarios can occur in real-world deployment of health machine learning applications. Users

---

[1]https://github.com/microsoft/RespireNet
[2]https://github.com/GuanLab/PDDB

| Health ML Models | In-Distribution | Out-of-Distribution | | |
|---|---|---|---|---|
| Skin Lesion (DenseNet-121) | HAM10000 92.05% | ISIC 2017 74.00% | Face N/A | CIFAR N/A |
| Lung Sound (ResNet-34) | ICBHI 2017 78.50% | Stethoscope 2.10% | AudioSet N/A | |
| Parkinson's Disease (5×1D-Conv) | mPower 82.01% | Kaggle Parkinson's 26.67% | MotionSense 45.83% | MHEALTH 10.00% |

Table 1: Accuracy of health deep learning models on in-distribution and out-of-distribution dataset. Accuracy is not available (N/A) for out-of-distribution datasets that do not have corresponding labels.

can input a face image to skin lesion classifier, improperly record lung sound and input ambient sound to the lung sound classifier, or input motion data when they are not at rest to the Parkinson's disease classifier. Furthermore, diverse sensors and devices used in real-world deployment can cause significant performance drop. This evaluation demonstrates that users are exposed to the health machine learning applications that can provide unreliable diagnostic results.

### 4.3 Out-of-Distribution Detection Performance

The previous evaluation implies that it is crucial to determine whether the input data belongs to in- or out-of-distribution to avoid failures caused by dataset shift. In this section, we investigate the feasibility of using state-of-the-art out-of-distribution detection methods in the context of machine learning for health. We evaluate Mahalanobis distance- [40], Gram matrices- [64], and energy-based [43] methods, which work on any pre-trained discrimitive classifiers and do not need re-training and prior knowledge of out-of-distribution datasets, in detecting out-of-distribution data for different health models.

#### 4.3.1 Experimental Setup

For Mahalanobis distance-based method[3], we extracted Mahalanobis distance-based scores from the output dense and residual block found in DenseNet and ResNet respectively. For the Parkinson's model which does not contain dense and residual blocks, we extracted the scores at the end of each convolutional layer. Then, we optimized the input noise magnitude using in-distribution samples and corresponding adversarial samples generated by FGSM [27]. The noise magnitude obtained is 0.0 for skin lesion classifier, 0.0005 for lung sound classifier, and 0.0 for Parkinson's disease classifier. For Gram matrices-based method[4], we extracted class-specific minimum and maximum correlation values for all orders of Gram matrices for all feature pairs. Total deviation values, which are used for out-of-distribution detection threshold, are computed with multiple sets of random samples from in-distribution datasets. For energy-based method[5], we use their method that does not require fine-tuning to avoid re-training of the network. We use the default temperature scaling ($T = 1$) as suggested in [43]. All evaluations are repeated for 5 trials and we report the mean (Table 2) and standard deviation (Table 4) of all metrics.

#### 4.3.2 Evaluation Metrics

For out-of-distribution detection, it is common to use true negative rate (TNR) at 95% true positive rate (TPR), AUROC, and detection accuracy to evaluate the performance of a detector. Particularly, as the out-of-distribution problem is a binary classification problem, we consider out-of-distribution samples as negative and in-distribution samples as positive. TNR at TPR 95% is defined as the percentage of correctly detected out-of-distribution samples, when 95% of in-distribution samples are correctly detected. The AUROC metric measures the area under the TPR vs FPR curve. The detection accuracy measures the maximum possible classification accuracy over all possible thresholds in distinguishing between in-distribution and out-of-distribution examples. Detailed explanations on the metrics are available in Appendix B.

---

[3] https://github.com/pokaxpoka/deep_Mahalanobis_detector
[4] https://github.com/VectorInstitute/gram-ood-detection
[5] https://github.com/wetliu/energy_ood

| Health ML Models | In-Distribution | Out-of-Distribution | Distribution Shift | Validation on OOD Samples (TNR @ TPR95/AUROC/Detection Accuracy) | | |
|---|---|---|---|---|---|---|
| | | | | **Mahalanobis** | **Gram** | **Energy** |
| Skin Lesion (DenseNet-121) | HAM10000 | ISIC 2017 | Covariate/label shift | 10.13 / 58.21 / 59.28 | 25.90 / 81.14 / 74.98 | 14.28 / 76.20 / 70.76 |
| | | Face | Open-set recognition | 100.00 / 99.98 / 99.96 | 95.01 / 98.20 / 96.34 | 0.00 / 80.45 / 84.81 |
| | | CIFAR10 | Open-set recognition | 99.83 / 99.90 / 99.61 | 95.14 / 98.66 / 96.90 | 5.06 / 58.33 / 57.89 |
| Lung Sound (ResNet-34) | ICBHI | AudioSet | Open-set recognition | 97.96 / 99.47 / 97.34 | 96.55 / 99.18 / 95.97 | 8.12 / 56.79 / 57.13 |
| | | Stethoscope | Covariate/label shift | 45.60 / 86.27 / 80.57 | 41.77 / 83.75 / 76.05 | 7.29 / 60.98 / 58.94 |
| Parkinson's Disease (5×1D-Conv) | mPower | MotionSense | Open-set recognition | 100.00 / 99.86 / 99.89 | 100.00 / 99.94 / 99.60 | 0.00 / 58.71 / 64.96 |
| | | mHealth | Open-set recognition | 100.00 / 100.00 / 100.00 | 100.0 / 99.99 / 99.99 | 0.00 / 41.41 / 59.44 |
| | | Kaggle Parkinson's | Covariate/label shift | 98.00 / 99.89 / 99.47 | 98.96 / 99.96 / 99.67 | 70.00 / 95.91 / 93.34 |

Table 2: Out-of-distribution detection performance across different networks and datasets.

### 4.3.3 Results

Table 2 shows out-of-distribution detection performance for different methods across different health machine learning models and datasets. Overall, Mahalanobis distance- and Gram matrices-based out-of-distribution detection methods consistently show outstanding performance across different neural networks and different out-of-distribution datasets, showing TNR @ TPR95 of 95% or above. These methods show lower performance in distinguishing near-distribution datasets (e.g., ISIC 2017, Stethoscope, Kaggle Parkinson's), which aligns with the results from previous out-of-distribution work [64]. On the other hand, the energy-based method did not show reasonable performance in detecting out-of-distribution samples. We found that the energy scores of out-of-distribution samples were not able to effectively discriminated from in-distribution samples as shown in Appendix C. Note that we used the energy scores without fine-tuning; however, the authors of energy score-based method [43] have demonstrated that a classifier that is fine-tuned using the energy score in place of the softmax score shows significant improvement in out-of-distribution detection performance. This evaluation implies that state-of-the-art out-of-distribution detectors can be applied to health machine learning applications to provide reliable diagnostic results to the users.

## 5 User Study

According to the trustworthy AI literature [20], providing users with interpretable information can enhance the trustworthiness of the result and potentially impact users' decisions. We therefore conducted an online survey-based user study to validate this effect and the impact that our approach has on model trustworthiness. We first defined CONFIDENCE SCORE as how confident the model is in interpreting an input. In other words, in-distribution input would have a high CONFIDENCE SCORE, whereas out-of-distribution input would have a low CONFIDENCE SCORE. We compute CONFIDENCE SCORE by scaling raw out-of-distribution scores from out-of-distribution detectors [40, 64] to 0–100, where 0 is most likely to be an out-of-distribution example and 100 is most likely to be an in-distribution example. Scaling is done in a piecewise manner. When out-of-distribution scores are within an in-distribution threshold, which is set to include 95% of in-distribution examples, we compute min-max scaling that ranges from 90 to 100. In this way, we ensure that most of the in-distribution examples have confidence scores of 90 or above. When out-of-distribution scores outside of an in-distribution threshold, we compute min-max scaling from 0 to 90, where the same denominator is used as above since out-of-distribution examples might not be available in practice and any negative values are clipped to 0. We then investigate the effect of CONFIDENCE SCORE on user-perceived trustworthiness and its impact on medical decisions. Additionally, we also quantify potential learning that can be gained when it comes to distinguishing between in- and out-of-distribution input samples. Specifically, we aim to answer the following research questions:

RQ. 1 How does the CONFIDENCE SCORE affect the perceived trustworthiness of diagnostic results?

RQ. 2 How does the CONFIDENCE SCORE affect medical decisions based on diagnostic results?

RQ. 3 Is there a potential learning effect from CONFIDENCE SCORE when it comes to distinguishing between input data with high and low CONFIDENCE SCORE?

### 5.1 Study Procedure and Participants

The overview of the online user study is illustrated in Figure 1 and a list of the user study interfaces is detailed in Appendix A. In short, the interface displays simulated results from the health screening models used in Section 4 (i.e., models for skin cancer, lung sound, and Parkinson's disease). We made

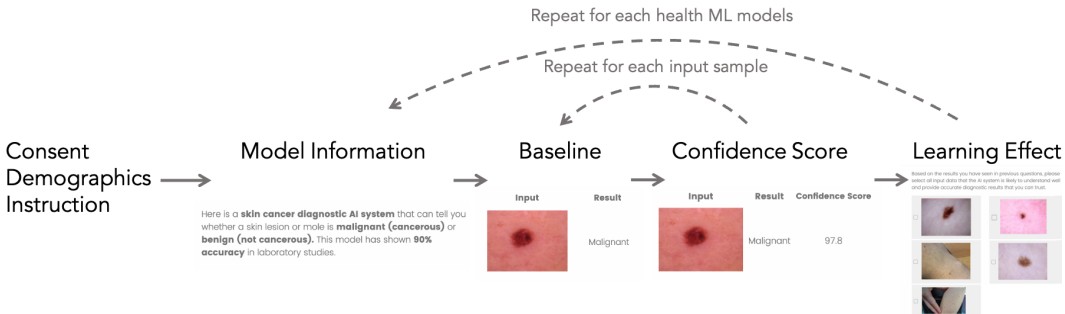

Figure 1: User study overview. The participants are first asked to give consent, read instruction, and provide demographics. Then, they report perceived trustworthiness and willingness to make a medical decision after seeing input samples that consist of different data types, diagnostic results, and CONFIDENCE SCORE for baseline and CONFIDENCE SCORE condition. Screenshots of the user study interface are demonstrated in Appendix A.

| | User-perceived trustworthiness | | | Impact on making medical decisions | | |
|---|---|---|---|---|---|---|
| | Wilcoxon Test ($W$) | $p$ | Effect Size ($r$) | Wilcoxon Test ($W$) | $p$ | Effect Size ($r$) |
| All | 529,950.0 | $< 0.001$*** | 0.393 | 223,227.0 | $< 0.001$*** | 0.178 |
| In-Distribution | 138,778.5 | $< 0.001$*** | 0.475 | 52,056.0 | $< 0.001$*** | 0.200 |
| Out-of-distribution | 126,814.0 | $< 0.001$*** | 0.317 | 59,790.5 | $0.001$*** | 0.158 |
| Negative result | 131,910.0 | $< 0.001$*** | 0.393 | 51,197.5 | $0.026$* | 0.100 |
| Positive result | 133,301.0 | $< 0.001$*** | 0.394 | 60,751.5 | $< 0.001$*** | 0.258 |
| Image | 55,890.0 | $< 0.001$*** | 0.436 | 20,884.0 | $< 0.001$*** | 0.225 |
| Audio | 64,440.0 | $< 0.001$*** | 0.384 | 25,848.0 | $0.002$** | 0.173 |
| Motion | 56,767.5 | $< 0.001$*** | 0.361 | 28,084.5 | $0.019$* | 0.133 |

Table 3: Results of Wilcoxon test in comparing baseline and CONFIDENCE SCORE conditions for the perceived trustworthiness and impact on decision making. All comparisons show statistically significant results. (***: $p < 0.001$, **: $p < 0.01$, *: $p < 0.05$).

the input data as human-readable as possible to maximize interpretability. Images were shown as is, while audio was included in an audio player so that participants could play, pause, and stop the track. We presented the motion data as a time-series plot of accelerometer signals from x-, y-, and z-axis. Since time-series data can be particularly challenging for non-experts, we explain that high-amplitude signals are associated with fast motion while while low-amplitude signals are associated with slow motion. The interface explained the model's purpose and accuracy, which was fixed to 90% to remove potential bias. For each model, the interface presents prediction results in two different conditions: (baseline) input and result, and (confidence score) input, result, and CONFIDENCE SCORE. For each result, we asked participants how much they trust the model's prediction and whether they would be willing to make a medical decision based on that result. Participants saw a total of 24 scenarios (3 data types (image, audio, motion) $\times$ 2 conditions (baseline vs. CONFIDENCE SCORE $\times$ 2 CONFIDENCE SCORE (high vs. low) $\times$ 2 results (positive vs. negative). To provide realistic experience, we provide different skin tone images for skin lesion samples based on the reported skin tone. With the exception of the data type, the scenarios were shuffled across all other factors to avoid any ordering effects. After participants saw all of the scenarios for a given data type, we presented them with five data examples and asked them to pick the ones that the model would be confident in processing according to CONFIDENCE SCORE. We added these questions to assess whether people were able to learn about how the CONFIDENCE SCORE was being generated after seeing a series of examples.

We intend to target ordinary, non-expert consumers at random rather than expert clinicians for our study. Our research is primarily directed toward the boom in consumer-facing mobile health applications, where non-expert users are expected to collect input data themselves. We believe this is where models are most susceptible to out-of-distribution inputs, providing unreliable predictions to the users. For the safe use of AI-powered health applications, the users would need support via automated uncertainty measures. To this end, We recruited participants from Amazon Mechanical Turk and compensated with $3 USD for a 15-min online study. In total, 192 participants (155 male, 67 female) completed the online study with an average age of $42.7 \pm 9.1$ years. The study was approved by Institutional Review Boards at the University of Washington.

## 5.2 Results

We analyzed the responses using the Wilcoxon signed-rank test [75] to compute a pairwise comparison of the categorical responses between the baseline and CONFIDENCE SCORE conditions. Table 3 summarizes these statistical results along with the Rosenthal correlation coefficient [62] ($r$) for effect size.

**RQ. 1: User-Perceived Trustworthiness** In general, we found that user-perceived trustworthiness ($p < 0.001$) was higher in the CONFIDENCE SCORE condition with medium effect size ($r = 0.393$). In other words, the dataset shift information helped the participants decide when to trust or not to trust the output of the models. Higher CONFIDENCE SCORES led to increasing trustworthiness; high scores had a large effect size ($r = 0.475$), while low scores had a medium effect size ($r = 0.317$). The impact on trustworthiness was similar for positive and negative diagnostic results. The effect sizes varied for the different input data types, with the images having the largest effect size and motion having the smallest. We suspect that the effect size was correlated with the intuitiveness of the data types, with images being more intuitive than motion data.

**RQ. 2: Impact on Making Medical Decisions** When we examined the impact of CONFIDENCE SCORE on making medical decisions, we found that there was a statistically significant difference ($p < 0.001$) between the baseline and CONFIDENCE SCORE conditions. In other words, participants were more willing to make medical decisions when positive results were presented with high CONFIDENCE SCORE and vice versa. Similar to the results for user-perceived trustworthiness, the effect of CONFIDENCE SCORE was larger on input data with high scores than low scores and highest for images compared to audio and motion data.

**RQ. 3: Learning Effect on Distinguishing In- and Out-of-Distribution Input Data** We found that the participants were able to learn from their interaction with CONFIDENCE SCORE. The average Jaccard index when it came to selecting high CONFIDENCE SCORE input data was 0.75, 0.66, and 0.64 for image, audio, and motion data, respectively, which is a moderately high similarity. As with our other results, the Jaccard index was highest on images and lowest on the motion data, implying that ability to understand the input data also has impact on learning effect. This implies that the dataset shift information can make users better understand input data that the machine learning models can interpret for the future interaction.

## 6 Discussion

**Dataset Shift Information for Health Application Users** Based on the results from our performance evaluation and user study, we can imagine two potential use cases of the dataset shift information to improve safety and trust in mHealth applications. First, mHealth applications with machine learning models can exclude out-of-distribution samples to avoid making inferences and suggestions that are likely to be inaccurate and unreasonable. Second, our user study shows that the dataset shift information can enhance their interaction with the health machine learning applications. It was found to be particularly effective in improving trustworthiness for in-distribution data and leading the users to make the right medical decision. As the users interact with the health applications longer, they would have better understanding of importance of data quality for future interactions.

**Dataset Shift Information for Health Application Developers** Our dataset shift information not only improves the user experience, but also yields potential benefits for model developers. If a user correctly captures data but the model rejects it as being out-of-distribution, then there likely exists intrinsic problems or biases with the model. For example, if a skin lesion classifier is only trained on data from people with pale skin and a user with darker skin submits an image of their own, the out-of-distribution detector be triggered due to the incompleteness of the training dataset. The same issues may occur when training dataset is only collected from a specific set of sensors (e.g., camera, microphone, IMU) with particular specifications.

**Limitations and Future Work** Detecting near-distribution samples (e.g., ISIC 2017, Stethoscope, Kaggle Parkinson's) was a difficult problem for all the out-of-distribution detectors we evaluated. For the near-distribution datasets, we evaluated model's accuracy on the data that are distinguished as in-distribution. This issue is actively investigated by researchers and the improved near-distribution detection method would benefit this work.

Our user study was limited in the fact that it dealt with hypothetical scenarios. There were no repercussions for users decisions, so they may not have spent as much time making their decisions as

they would in real life. There are also many other factors that impact people's health-related decision making, such as the perceived severity of the medical condition and the perceived benefits of taking action [33, 67]. We tried to make some of the data more realistic by aligning data with the user's demographic information (e.g., we displayed skin lesion images based on their reported skin tone); nevertheless, participants were aware that the data was not their own. Additionally, increased trust might be affected by participants' own understanding and interpretation of the input data. Although we observed increased and decreased trust in examples with high and low CONFIDENCE SCORE, respectively, randomizing the CONFIDENCE SCORE of input data could further quantify impact of CONFIDENCE SCORE on user trust of health predictions.

In future work, we would like to (1) investigate the best way to present this information for the users, (2) leverage the dataset shift information for finding potential biases in the train dataset and inherent problems with the model, (3) investigate an out-of-distribution method for better near-distribution detection performance.

## 7    Conclusion

In this work, we investigated the utility of dataset shift information for improving reliability and trustworthiness of machine learning-based health applications. Using publicly available health deep learning models and datasets, we first demonstrated that the models fail when encountered with unseen data. We then evaluated the out-of-distribution detection performance of state-of-the-art methods, showing high accuracy in distinguishing between in- and out-of-distribution datasets for different input data types (e.g., image, audio, motion data). We conducted an online user study to investigate the effect of dataset shift information on potential users. We found that the participants trusted prediction results with high CONFIDENCE SCORE and are more willing to make a right medical decision, while they considered prediction results with low CONFIDENCE SCORE less trustworthy and are less willing to make medical decision. This work shows that the dataset shift is a meaningful piece of information for building consumer-facing trustworthy AI applications for high-stake decision making.

## 8    Acknowledgments and Disclosure of Funding

We would like to thank Alex Mariakakis for insightful feedback and discussions. This work was supported by NSF CNS-1565252. Shwetak Patel is on partial leave at Google.

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
