## A  User Study Interface

In this section, we provide screenshots and list of examples that were used in the user study.

We are conducting a research study to better understand the acceptability of AI-based system that can aid diagnostic medical screening. We estimate this online study will take approximately 15 minutes to complete. Please answer each question as completely and honestly as you can.

As compensation for your participation, you will be paid with $3. At the end of the survey, an ID number will be provided for you to paste into MTurk.

There is no risk to participating in this study, and you may withdraw from the study at any time. All of the information will be confidential, only accessible by approved research collaborators. The data will be used to guide the design of our future research. The emails and addresses will be kept in a list separate from and not connected to the data.

If you have any questions or concerns, please contact:
- mhealth-survey@cs.washington.edu

If you would like to talk to someone separate from the research team about a concern or complaint about your rights as a possible research subject, please contact the University of Washington Institutional Review Board at (206) 543-0098. We cannot ensure the confidentiality of any information sent by email. This study has been approved by the University of Washington's Human Subjects Division under IRB Study #STUDY00013036.

By clicking "I agree", you agree:
- That you are at least 18 years of age,
- That you do not have impaired vision and/or hearing,
- That you are participating in this study, and
- That you understand you can withdraw from the survey at any time,

⚪ I agree

⚪ Leave

Figure 2: User study consent form. Note that the name of the institution is redacted for the review.

Here is a **skin cancer diagnostic AI system** that can tell you whether a skin lesion or mole is **malignant (cancerous)** or **benign (not cancerous).** This model has shown **90% accuracy** in laboratory studies.

(a) Interface that shows information about a health machine learning model. It shows target health condition, possible prediction results, and its accuracy.

The AI system now shows you the diagnostic result with additional information, "**Confidence Score.**"

**Confident Score**: This score shows how confident the AI system in **understanding your input data**. The score ranges from 0 to 100.

**100** is when the AI system is **most confident** in understanding the input; it is highly likely that the AI system **has seen similar data** when the system is being developed.

**0** is when the AI system is **least confident** in understand the input data; it is highly likely that the AI system **has never seen similar data** when the system is being developed.

| Input | Result | Confidence Score |
|---|---|---|
| | Malignant | 99.7 |

*Confident score ranges from 0 to 100. **0**: AI system doesn't understand the input. **100**: AI system understands the input.

How much do you trust the AI system's diagnostic result?

○ Extremely
○ Very much
○ Moderately
○ Slightly
○ Not at all

Would you decide to go see a doctor after seeing on this result?

○ Yes
○ Maybe
○ No

(c) Interface that shows CONFIDENCE SCORE condition. This condition only presents input data, prediction results, and CONFIDENCE SCORE.

Imagine you provide the below image to the AI diagnostic system. And, the AI system shows you the below information.

**Input** is the input data that you provided to the AI system.
**Result** is the diagnostic result provided by the AI system.

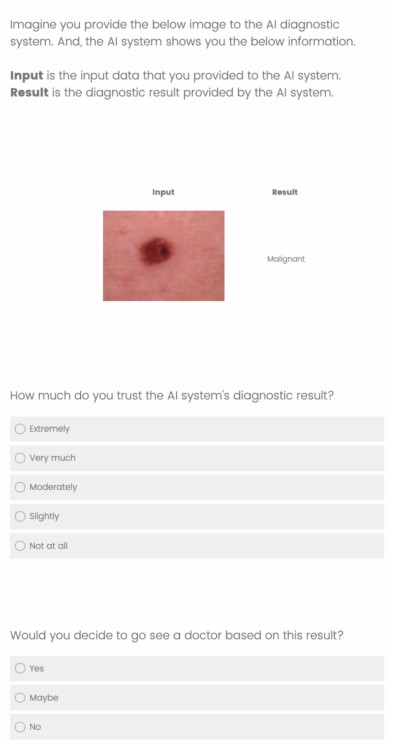

Input    Result

Malignant

How much do you trust the AI system's diagnostic result?

○ Extremely
○ Very much
○ Moderately
○ Slightly
○ Not at all

Would you decide to go see a doctor based on this result?

○ Yes
○ Maybe
○ No

(b) Interface that shows baseline condition. This condition only presents input data and prediction results and asks questions on user-perceived trustworthiness and impact on making medical decisions.

Based on the results you have seen in previous questions, please select all input data that the AI system is likely to understand well and provide accurate diagnostic results that you can trust.

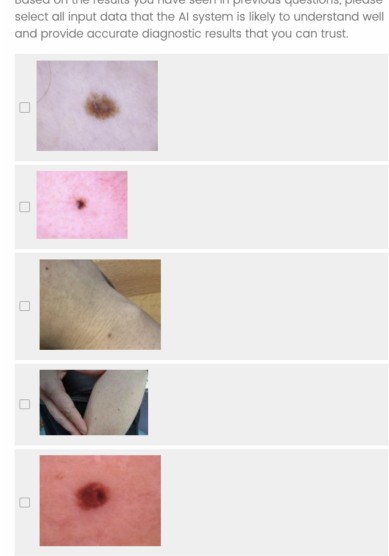

(d) Interface that asks users to select input data that would have high CONFIDENCE SCORE to explore potential learning effect through their interaction with CONFIDENCE SCORE.

Figure 3: List of user study interface. This shows an example interface for skin cancer classifier.

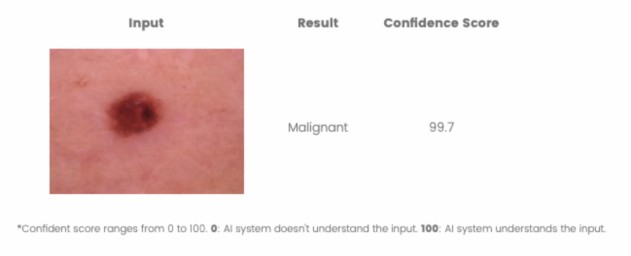

(a) Image input is shown in a visible size.

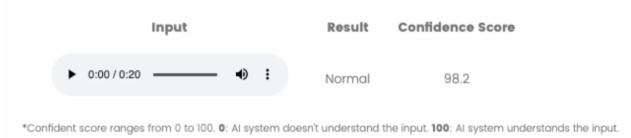

(b) Audio player is embedded for the participants to listen to the input data.

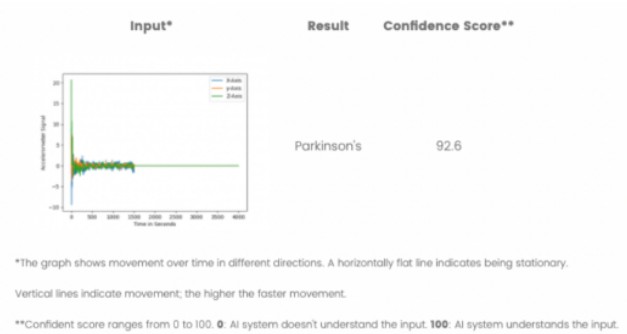

(c) Motion data is shown as a time-series plot of accelerometer signal. We provide additional explanation about how to interpret the signal.

Figure 4: Interface to display different input data types.

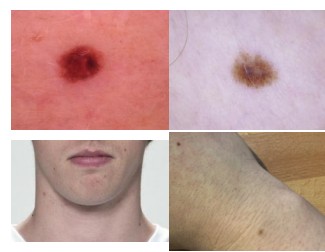

(a) Input examples for skin cancer classifier for the participants who self-report to have light-colored skin tone.

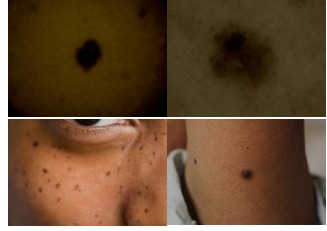

(b) Input examples for skin cancer classifier for the participants who self-report to have dark-colored skin tone.

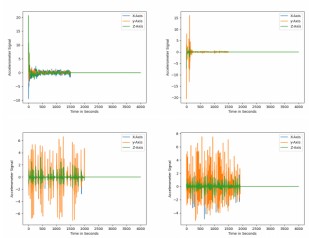

(c) Input examples for Parkinson's disease classifier.

Figure 5: List of input examples used in the user study. For each input type, top row shows in-distribution inputs and bottom row shows out-of-distribution inputs. Left column shows positive diagnostic results and right column shows negative diagnostic results. Note that audio samples are not included due to its difficulty to visualize.

# B  Performance Metrics

In out-of-distribution performance evaluation in Section 4.3, we use the following metrics that has been used in previous out-of-distribution work [40, 64]:

- **True negative rate (TNR) at 95% true positive rate** (TPR) is defined as the percentage of correctly detected out-of-distribution samples, when 95% of in-distribution samples are correctly detected. TNR is calculated $TNR = TN/(FP + TN)$ and $TPR = TP/(TP + TN)$, where TP, TN, FP, and FN are true positive, true negative, false positive, and false negative, respectively.

- **Area under the receiver operating curve (AUROC)** is defined as the area under the plot of true positive rate (TPR) versue false positive rate (FPR), where $TPR = TP/(TP + FN)$ and $FPR = FP/(FP + TN)$.

- **Detection accuracy** is defined as the maximum classification accuracy over all possible thresholds in classifying in- and out-of-distribution data.

# C  Energy-Based OOD Detection Analysis

In Section 4.3, energy-based out-of-distribution detection method does not show comparable performance to methods using Mahalanobis distance and Gram matrices. We further analyze the method by comparing the distribution of energy score between in- and out-of-distribution as shown Figure 6. In most cases, the distribution of the energy scores are overlapped, making it difficulty to detect out-of-distribution samples using energy score. In this work, we use energy-based method without fine-tuning, which is suitable for adopting the method to any pre-trained models. However, as the authors have demonstrated in their paper [43], fine-tuned energy-based method that requires re-training of a classifier, shows significant improvement in detecting out-of-distribution samples.

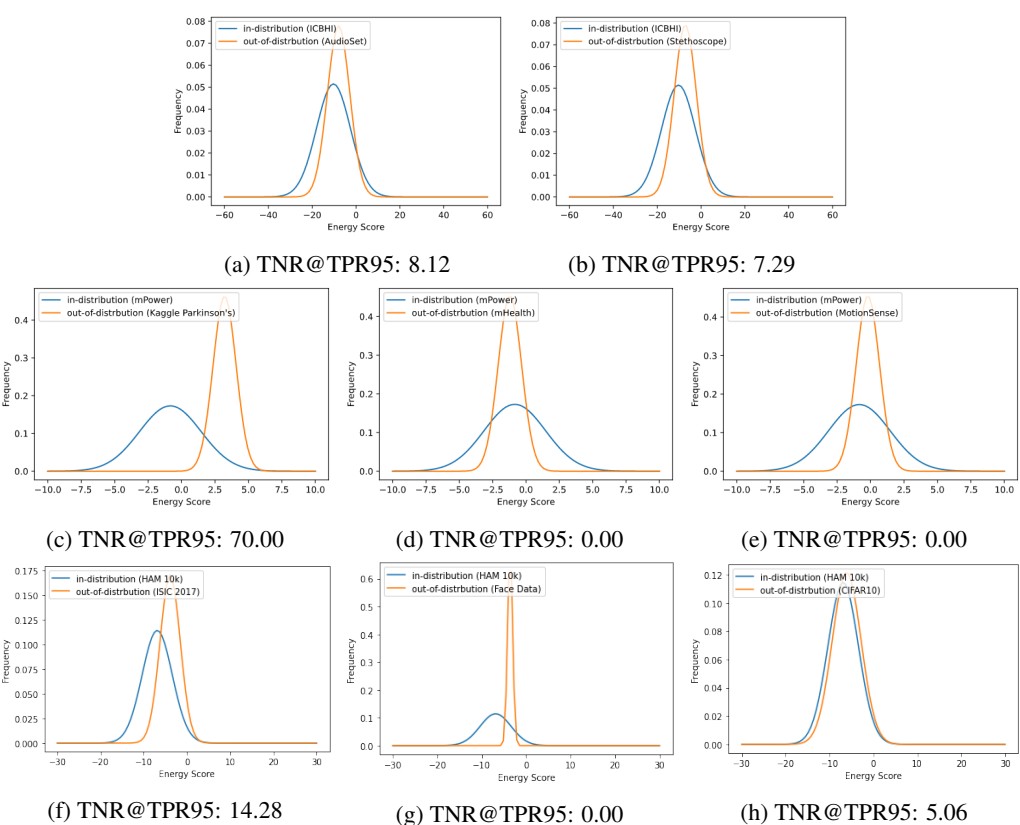

Figure 6: Energy score distribution across different in- and out-of-distribution datasets.

## C.1 Out-of-Distribution Performance with Confidence Interval

| Health ML Models | In-Distribution | Out-of-Distribution | Distribution Shift | Validation on OOD Samples (TNR @ TPR95/AUROC/Detection Accuracy) | | |
|---|---|---|---|---|---|---|
| | | | | **Mahalanobis** | **Gram** | **Energy** |
| Skin Lesion (DenseNet-121) | HAM10000 | ISIC 2017 | Covariate/label shift | 10.13 / 58.21 / 59.28 ±2.61/ ±3.30 / ±2.38 | 25.90 / 81.14 / 74.98 ±1.22 / ±1.89 / ±1.12 | 14.28 / 76.20 / 70.76 ±0.49 / ±0.18 / ±0.16 |
| | | Face | Open-set recognition | 100.00 / 99.98 / 99.96 ±0.00 / ±0.02 / ±0.04 | 95.01 / 98.20 / 96.34 ±1.48 / ±0.41 / ±0.63 | 0.00 / 80.45 / 84.81 ±0.00 / ±0.14 / ±0.25 |
| | | CIFAR10 | Open-set recognition | 99.83 / 99.90 / 99.61 ±0.18 / ±0.10 / ±0.39 | 95.14 / 98.66 / 96.90 ±1.43 / ±1.37 / ±1.94 | 5.06 / 58.33 / 57.89 ±0.26 / ±0.92 / ±0.67 |
| Lung Sound (ResNet-34) | ICBHI | AudioSet | Open-set recognition | 97.96 / 99.47 / 97.34 ±0.73 / ±0.26 / ±0.45 | 96.55 / 99.18 / 95.97 ±1.67 / ±0.30 / ±0.62 | 8.12 / 56.79 / 57.13 ±0.24 / ±0.15 / ±0.14 |
| | | Stethoscope | Covariate/label shift | 45.60 / 86.27 / 80.57 ±4.95 / ±1.42 / ±1.55 | 41.77 / 83.75 / 76.05 ±1.62 / ±0.63 / ±0.38 | 7.29 / 60.98 / 58.94 ±1.22 / ±0.74 / ±0.63 |
| Parkinson's Disease (5×1D-Conv) | mPower | MotionSense | Open-set recognition | 100.00 / 99.86 / 99.89 ±0.00 / ± 0.13 / ±0.10 | 100.00 / 99.94 / 99.60 ±0.00 / ±0.24 / ±0.14 | 0.00 / 58.71 / 64.96 ±0.00 / ±0.59 / ±0.32 |
| | | mHealth | Open-set recognition | 100.00 / 100.00 / 100.00 ± 0.00 / ±0.00 / ±0.00 | 100.0 / 99.99 / 99.99 ±0.00 / ±0.02 / ±0.01 | 0.00 / 41.41 / 59.44 ±0.00 / ±1.09 / ±1.10 |
| | | Kaggle Parkinson's | Covariate/label shift | 98.00 / 99.89 / 99.47 ±2.45 / ±0.14 / ±1.25 | 98.96 / 99.96 / 99.67 ±0.00 / ±0.02 / ±0.03 | 70.00 / 95.91 / 93.34 ±4.68 / ±0.30 / ±0.32 |

Table 4: Out-of-Distribution Detection Performance Across Multiple Tasks. Evaluation is repeated for 5 times. Mean and standard deviation of metrics are reported.