# OpenReview forum: "Reliable and Trustworthy Machine Learning for Health Using Dataset Shift Detection"
_NeurIPS.cc/2021/Conference — NeurIPS 2021 Poster_

### Official Review · Reviewer_QZBK · 2021-07-06

**Rating:** 6
**Confidence:** 5

**Summary:**

This work investigates the effect of out-of-distribution data on machine learning models on healthcare applications. The authors first display a decrease in model performance on near and far out-of-distribution samples for 3 health applications. They then assess the performance of 3 techniques for out-of-distribution detection and build a confidence score from this binary classifier. The confidence score is assessed in a user-study as a complementary piece of information to be presented to the user.

**Ethical Concerns:**

Out of curiosity: is $3 for 15 minutes a normal pay? 12$/ hour seems to be below the minimum pay in the US (at least in some states), and is low for European standards as well.

**Limitations And Societal Impact:**

The authors acknowledge the 'theoretical' setup of the healthcare applications. This remains a limitation, as it is unclear whether predictions in themselves are useful in clinical settings. However, I think that the proposed framework has merits in terms of ood detection and transparency with the user, that can lead to more realistic studies.

I would encourage the authors to highlight a bit more the user study and discuss its limitations.

**Main Review:**

Originality
---------------
Out-of-distribution detection is not a novel problem, and technically the methods proposed are not new. The authors mention that their application to healthcare data makes this piece different, however I found no evidence that the considered healthcare applications affected the techniques or their use. Similarly, a decrease of model performance on ood data has been highlighted in several works (see works referring to 'robustness' to e.g. new diseases, time shifts or local perturbations). On the other hand, user studies are rarely conducted when claiming better ood detection methods. I would therefore encourage the authors to highlight the user-study more.

Quality
----------
The work is overall well executed and convicing. I have a couple of questions for the authors that would help me to better assess the impact, especially of the user study:
- In the Appendix, the questions refer to the user's trust in the system, and whether they would see a doctor if given this diagnosis. Is the assumption that there is a deterministic relationship between the model's prediction of e.g. malignancy with whether they would consult a doctor? Can a decision be factorized into 2 components: how the user/patient combines the model's prediction (+ confidence score) with their own priors to make a diagnosis themselves, and whether they decide to see a doctor based on this diagnosis? Or is it assumed that the model is only informational and that the user/patient task is to decide whether they see a doctor or not? Was there any investigation into which mental model of the classifier the user were referring to? If the former, it could have been interesting to add a question about what their diagnosis was. I understand this is probably a broader issue about the perception of a classifier as a diagnosis vs referral vs informational tool, but I was wondering whether the authors had any thoughts?
- On the increased user trust: Is the additional information increasing trust, independently of its quality? One way to test this would be to provide randomized confidence scores to images, to understand whether the added trust comes from the quality of the score and the correct understanding of the user, or whether it comes from a perception of increased information.

Clarity
---------
The paper is well written and easy to follow.

Significance
-----------------
I believe this is a nice application of existing techniques, well executed. I think the most interesting part is the user study, and I think it could be improved to have even more impact (see questions above).


References:
- Does Your Dermatology Classifier Know What It Doesn’t Know? Detecting the Long-Tail of Unseen Conditions https://arxiv.org/pdf/2104.03829.pdf
- Feature Robustness in Non-stationary Health Records, http://proceedings.mlr.press/v106/nestor19a/nestor19a.pdf

**Time Spent Reviewing:**

2.5

---

> ### Author Response · Authors · 2021-08-10
> **Authors' Response**
>
> We would like to thank the reviewer for their positive comments on execution of the work and the user study. We also appreciate detailed feedback on user study design and suggestions for improvement. We agree that the user study shows meaningful insights on the user's perception on ML-based health applications. As suggested by the reviewer, we will highlight the user study for the camera-ready version. Below, we address questions raised by the reviewers to improve our paper.
>
>
> **Relevance to healthcare applications**
>
> We agree that we have not designed a new OOD detection method specific for healthcare applications. Our paper focuses on utilizing the existing state-of-the-art OOD detection methods for healthcare applications. However, most OOD detection methods are built and evaluated on image datasets. Audio and accelerometer data also contain useful information about people’s pulmonary or neurological health conditions, yet the performance of OOD detection methods on such has been unexplored. To the best of our knowledge, this is the first paper that extends the OOD evaluation to these sensing modalities.
>
>
> **Relationship between health prediction and medical decision making**
>
> The Health Belief Model (HBM) [1, 2] is one of many psychological frameworks that have been proposed to explain and predict health-related decision-making. In short, the model suggests the likelihood that someone will take a health-promoting action (e.g., visiting the doctor) depends on the four following constructs: the individual’s perceived susceptibility to the condition, their perception of the condition’s severity, the perceived benefits of taking action, and the potential barriers that may prevent them from taking action. Showing participants results from diagnostic tests is one way of increasing their perceived susceptibility to the condition, and our hypothesis was that we could amplify that effect further by including a confidence score that increased their trust in the model. Although it would have been possible to ask people to rate their perceived susceptibility, we decided to focus on the distal outcome of taking action for a couple of reasons: (1) taking action is a concrete binary outcome that people have an easier time conceptualizing and articulating than perceived susceptibility; and (2) since the goal of many mobile health apps is to compel consumers to take action about their health, we felt it would be more compelling to see if increasing trust in the model had a significant enough change to affect such a distal outcome. Nevertheless, we believe that future work could expand on our user study by decomposing health-related decision-making into the aforementioned components.
>
>
> **Trustworthiness improvement from additional information vs confidence score**
>
> We agree that the presence of additional information could contribute to increase in trust. However, from the user study results, we also observed increased trust in examples with high confidence scores and decreased trust in examples with low confidence scores. This means that the users were able to decide when to trust or not trust predictions by interpreting the confidence scores in a logical manner. Nevertheless, we appreciate the idea of randomizing the images’ scores to further quantify the impact of additional information versus confidence scores, so we will include this point in the discussion section.
>
>
> **'Theoretical' setup of the healthcare applications**
>
> We believe the experimental setting represents diverse mobile health scenarios. In recent years, various mobile health applications have been made to consumers, such as Apple’s atrial fibrillation detection [3], Google’s dermatology tool [4], respiratory/heart rate detection [5], and sleep tracking [6]. These mobile health applications and the ones we included in our paper are not designed to provide official clinical diagnosis but rather to serve one of two purposes: (1) early screening for conditions that have not yet been diagnosed so that people can be prompted to go to their healthcare provider for a more rigorous assessment, or (2) continuous monitoring for conditions that have already been diagnosed. For mobile health applications, non-expert users are expected to input data collected by themselves. We believe this is where models are most susceptible to out-of-distribution inputs, providing unreliable predictions to the users. For example, when users are asked to input a close-up of a mole, they might input an image of a face where moles are barely visible and a skin lesion classifier provides an unreasonable prediction. Unlike clinicians, who may either receive training on how to operate these mobile apps or may already understand what must be done to generate high-quality, in-distribution data, we believe non-expert consumers require support via automated uncertainty measures.
>
>
> **Turker compensation**
>
> We thank the reviewer for bringing up important ethical concerns. We made sure that the participants are paid above the US federal minimum wage of 7.25 USD/hr and the average state minimum wage of 9.48 USD/hr. Hara et al. [7] showed that average Mechanical Turk requesters pay around 11 USD/hr. In the mTurk worker community [8, 9], it is generally accepted that the workers expect compensation between 10 USD/hr and 12 USD/hr.
>
> [1] Siddiqui, Taranum Ruba, et al. "Use of the health belief model for the assessment of public knowledge and household preventive practices in Karachi, Pakistan, a dengue-endemic city." PLoS neglected tropical diseases 10.11 (2016): e0005129.
>
> [2] Janz, Nancy K., and Marshall H. Becker. "The health belief model: A decade later." Health education quarterly 11.1 (1984): 1-47.
>
> [3] Heart health notifications on your Apple Watch. Apple Support. (2021, April 26). https://support.apple.com/en-us/HT208931.
>
> [4] Bui, P. (2021, May 18). Using ai to help find answers to common skin conditions. Google. https://blog.google/technology/health/ai-dermatology-preview-io-2021/.
>
> [5] Patel, S. (2021, February 4). Take a pulse on health and wellness with your phone. Google. https://blog.google/technology/health/take-pulse-health-and-wellness-your-phone.
>
> [6] McHugh-Johnson, M. (2021, March 30). Sleeping on the job: How we built the new Nest hub. Google. https://blog.google/products/google-nest/sleeping-job-how-we-built-new-nest-hub/.
>
> [7] Hara, Kotaro, et al. "A data-driven analysis of workers' earnings on Amazon Mechanical Turk." Proceedings of the 2018 CHI conference on human factors in computing systems. 2018.
>
> [8] how to figure out a fair wage? (2021, February 1). Reddit. https://www.reddit.com/r/mturk/comments/la4oq4/how_to_figure_out_a_fair_wage/
>
> [9] TurkerView | Worker Review Platform for Amazon Mechanical Turk. (n.d.). TurkerView. Retrieved August 9, 2021, from https://turkerview.com/

---

### Official Review · Reviewer_hRAn · 2021-07-16

**Rating:** 6
**Confidence:** 4

**Summary:**

The authors provide a small scale study of out of distribution detection methods on three medical datasets and perform a survey of Turkers to see how these OODs impact trustworthiness and medical decision making. They find that Gram and Manhabolis based methods detect OOD data well and that these scores translate into increased trustworthiness and medical decision making in their survey.

**Limitations And Societal Impact:**

Yes. This research involved human subjects and the authors state they received appropriate IRB approval for it.

**Main Review:**

### Main comments:
I would like to commend the author's for taking the time to conduct a survey to see how their experimental results translate into user perception and behavior. A very rare thing for computational papers!

In addition to my detail comments below, my primary concern about this paper stems from the scope of its evaluations. 3 datasets and 3 methods is good, but still modest by most standards. Furthermore, the "difficulty" of the OOD scenarios tested is unlikely to be reflective of real clinical scenarios. Realistic OOD in healthcare is partially captured by some of the examples, but a practitioner in this space will likely still have many questions about which if any OOD detection methods will work in practice. Much more subtle forms of distributional shift occur in practice and inclusion of these would strengthen the conclusions.

I also think that the author's should more formally define what they mean by OOD, i.e. are they testing covariate shift, label shift, or something else? Providing this additional specificity will help users know if the authors' results are applicable to their wrork.

### Detailed comments:

__Originality__: The work is original and includes both a small computational evaluation and user survey.

__Quality__:  The experimental sections are sound and the survey is as well done, if a bit small and artificial. It would be great to include the distribution of softmax probabilities on the OOD data. I am curious to know if the models are extremely confident on data that has nothing to do with their training task.

- I do not believe _CONFIDENCE SCORE_ is ever formally define. Please include this definition.

__Clarity__: The work is clearly presented and I found text and experiments easy to follow.

__Significance__: The work looks at a very important topic that is understudied. Further experiments would increase the significance of this work.

**Time Spent Reviewing:**

2

---

> ### Author Response · Authors · 2021-08-10
> **Authors' Response**
>
> We would like to thank the reviewer for their positive comments on our work and recognizing the importance of the problem we aim to address in the paper. We also appreciate the comment on our user study. We hope to share our findings on the user's perception on ML-based health applications with the NeurIPS community. Below, we address questions raised by the reviewers to improve our paper.
>
> **Definition of OOD**
>
> We admit that OOD is a broadly used term. As mentioned in the paper, we aim to include diverse scenarios caused by dataset shift. Specifically, we aim to include OOD datasets caused by covariate shift, label shift, and open-set recognition. For the skin lesion classifier, ISIC-2017 represents both covariate and label shift by containing skin lesion images from a different source and having a different label distribution. Meanwhile, Face/CIFAR represents open-set recognition by including classes of images that are not used during training. For the lung sound classifier, Stethoscope represents covariate and label shift, while AudioSet represents open-set recognition. For the Parkinson’s classifier, Kaggle Parkinson’s represents covariate and label shift, while MotionSense and mHealth represent open-set recognition. For the camera-ready version, we will add a column in Table 2 to explain what each dataset represents in terms of dataset shift type.
>
> **Realistic OOD scenarios**
>
> We thank the reviewer for pointing out a critical aspect of the experiment setting. In this work, we aim to include both near-distribution datasets and out-of-distribution datasets for the OOD evaluation. Near-distribution datasets contain similar data with in-distribution, but collected in a different setting or using a slightly different device. For example, OOD detection for a lung sound classifier trained on ICBHI is evaluated with Stethoscope, both of which are lung sound datasets. For skin cancer and Parkinson's classifier, near-distribution datasets are ISIC2017 and Kaggle Parkinson's, respectively. We found that the existing OOD detection methods show limited accuracy for these near-distribution scenarios, which is discussed in the limitation section.
>
> We demonstrated that the OOD detection on OOD datasets shows extremely high accuracy. We agree that the OOD examples can be distinguishable by users who have expertise in the clinical and medical field. However, as consumer-facing mobile health applications become more widely available, so too is the need for providing uncertainty measures. For example, when users are asked to input a close-up of a mole, they might input an image of a face where moles are barely visible and a skin lesion classifier provides an unreasonable prediction. Unlike clinicians and other domain experts, non-expert consumers may not have sufficient expertise to distinguish even the obvious examples of OOD data. In that sense, we believe the experimental setting represents realistic consumer-facing mobile health scenarios.
>
> **Softmax probabilities on OOD examples**
>
> This phenomenon is well studied in previous work [1, 2], showing that deep neural networks fail on OOD examples with very high softmax probabilities; we observed the same phenomenon during our experiments. For the camera-ready version, we will include the distributions of softmax probabilities for our OOD datasets in the Appendix.
>
> **Confidence score definition**
>
> Our confidence score is based on the Mahalanobis OOD score since it had the most reliable OOD detection performance in our experiments. To translate the OOC scores into human-interpretable confidence score, we then scaled those scores from 0 to 100; 0 being the most likely to be an OOD example and 100 being the most likely to be an in-distribution example. Scaling is done in a piecewise manner. When OOD scores are within an in-distribution threshold, which is set to include 95% of in-distribution (i.e., train set) examples, we compute min-max scaling that ranges from 90 to 100, where min and max are the minimum OOD score of in-distribution examples and the threshold, respectively. In this way, we ensure that most of the in-distribution examples have confidence scores of 90 or above. When OOD scores outside of an in-distribution threshold, we compute min-max scaling from 0 to 90, where the same denominator is used as above since OOD examples might not be available in practice and any negative values are clipped to 0. We believe confidence score used in this work is generalizable to other OOD methods that output OOD score or probability of example being OOD. We will add the definition of confidence score used in the user study for the camera-ready version.
>
> [1] Liu, Jeremiah Zhe, et al. "Simple and principled uncertainty estimation with deterministic deep learning via distance awareness." arXiv preprint arXiv:2006.10108 (2020).
>
> [2] Nguyen, Anh, Jason Yosinski, and Jeff Clune. "Deep neural networks are easily fooled: High confidence predictions for unrecognizable images." Proceedings of the IEEE conference on computer vision and pattern recognition. 2015.

---

### Official Review · Reviewer_Ku9s · 2021-07-17

**Rating:** 4
**Confidence:** 5

**Summary:**

This work applies known out-of-distribution (OOD) detection techniques on various medical datasets and confirms that the OOD detection techniques indeed works. Specifically, the medical prediction models tend to make more incorrect predictions for samples that OOD detection algorithms deem OOD.

**Main Review:**

Pros
- This paper focuses on an important problem; the reliability of prediction models in healthcare. They specifically employ OOD detection methods to test if they can help increase the trustworthyness of prediction models.
- The authors conduct extensive experiments using medical datasets of multiple modalities (e.g. images, audio) and multiple conditions (e.g. skin cancer, lung sound) to provide reliable conclusion

Cons
- The focus of this work is to study whether the existing OOD detection methods can work in various medical datasets. Therefore there is limited methodological novelty, which is the main interest of the NeurIPS community.

Summary
- The work itself is valuable, but probably not in NeurIPS. This work would be much more appreciated in relevant venues just such as MLHC, CHIL or JAMIA, JMIR.

Post Rebuttal
- I appreciate the authors taking the time to describe their thoughts in detail. However, I do not belive that simply applying well-known OOD detection methods on publicly available datasets justifies an acceptance to this venue, as that is not the main interest of this community. Therefore I'll keep my initial rating. (The user study is an interesting aspect, and will be of huge relevance to the healthcare community)

**Time Spent Reviewing:**

2

---

> ### Author Response · Authors · 2021-08-10
> **Authors' Response**
>
> We appreciate the critical feedback the reviewer has provided on the methodological novelty of our work and its fit to the NeurIPS community. Although we agree that MLHC, CHIL, JAMIA, and JMIR could potentially be all good fits for this work, we also believe that the findings from this work are valuable to share with the NeurIPS audience as it touches on both computational contributions and the user experience of ML applications for healthcare. Our work specifically addresses “Social Aspects of Machine Learning”, which is the last bullet within the Call for Papers [1]. As ML applications for healthcare become increasingly available to the non-expert consumers, it is crucial that the community explores various ways of conveying reliability and trustworthiness when users are asked to make high-stake medical decisions.
>
> Methodologically, one of our key motivations was to investigate a technique that could generalize across sensing modalities within the umbrella of mobile health. Most OOD detection methods  have been created and tested on image datasets. To the best of our knowledge, our work is the first paper to extend OOD detection to audio and accelerometer datasets. This is important in health applications where audio and accelerometer data contain useful information about people’s pulmonary or neurological health conditions. This work further investigates the efficacy of OOD detection on improving user-perceived trustworthiness through a user study. We believe that insights from the user study are novel and compelling for the NeurIPS community, as the other reviewers have commented that it is rare for computational papers.
>
> [1] Neural Information Processing Systems. (n.d.). NeurIPS 2021 call for papers. https://neurips.cc/Conferences/2021/CallForPapers.

---

### Official Review · Reviewer_hXoK · 2021-07-19

**Rating:** 6
**Confidence:** 4

**Summary:**

This paper touches on a very interesting and important problem in the applied machine learning field, the trustworthiness of ML models, how to measure it, and its effect on user experience. The paper presents a very nice introduction to the topic and opens the problem in the context of out-of-distribution detection. Three different approaches for out-of-distribution detection are benchmarked on several datasets and ML methods. Furthermore, a confidence score is proposed to translate the out-of-distribution scores into more interpretable scores for the users. A user study is further conducted to evaluate the effect of using confidence scores on the trust level of human participants.

**Main Review:**

The paper discusses a significant problem in the field and the text is very clear. I have few concerns regarding the experimental setups and results:
1) It seems from the text that the user study was performed on random participants. In order to truly measure the effect of using the confidence score on the perceived trustworthiness, the experiments must be performed on expert participants (as true users of the ML model) with domain knowledge about the experimental data.
2) It is not clear from the text, how the confidence score is computed.
3) The authors may also discuss the advantage of their proposed confidence score with commonly used metrics in the field such as uncertainty measures.


**Time Spent Reviewing:**

2

---

> ### Author Response · Authors · 2021-08-10
> **Authors' Response**
>
> We would like to thank the reviewer for their insightful comments and recognizing the merits of our work. We agree that our work attempts to address an important problem of building trustworthy and reliable machine learning models for high-staking decision making, such as healthcare. Below, we address questions raised by the reviewers to improve our paper.
>
> **Random participants vs expert participants**
>
> We intended to target ordinary consumers at random rather than expert clinicians for our study. Our research is primarily directed towards the boom in consumer-facing mobile health applications, such as Apple’s atrial fibrillation detection [1], Google’s dermatology tool [2], respiratory/heart rate detection [3], and sleep tracking [4]. These mobile health applications and the ones we included in our paper are not designed to provide official clinical diagnosis but rather to serve one of two purposes: (1) early screening for conditions that have not yet been diagnosed so that people can be prompted to go to their healthcare provider for a more rigorous assessment, or (2) continuous monitoring for conditions that have already been diagnosed. For mobile health applications, non-expert users are expected to input data collected by themselves. We believe this is where models are most susceptible to out-of-distribution inputs, providing unreliable predictions to the users. For example, when users are asked to input a close-up of a mole, they might input an image of a face where moles are barely visible and a skin lesion classifier provides an unreasonable prediction. Unlike clinicians, who may either receive training on how to operate these mobile apps or may already understand what must be done to generate high-quality, in-distribution data, we consider consumers to be non-experts who require support via automated uncertainty measures.
>
>
> **Confidence score definition**
>
> Our confidence score is based on the Mahalanobis OOD score since it had the most reliable OOD detection performance in our experiments. To translate the OOC scores into human-interpretable confidence score, we then scaled those scores from 0 to 100; 0 being the most likely to be an OOD example and 100 being the most likely to be an in-distribution example. Scaling is done in a piecewise manner. When OOD scores are within an in-distribution threshold, which is set to include 95% of in-distribution (i.e., train set) examples, we compute min-max scaling that ranges from 90 to 100, where min and max are the minimum OOD score of in-distribution examples and the threshold, respectively. In this way, we ensure that most of the in-distribution examples have confidence scores of 90 or above. When OOD scores outside of an in-distribution threshold, we compute min-max scaling from 0 to 90, where the same denominator is used as above since OOD examples might not be available in practice and any negative values are clipped to 0. We believe confidence score used in this work is generalizable to other OOD methods that output OOD score or probability of example being OOD. We will add the definition of confidence score used in the user study for the camera-ready version.
>
> **Comparison to other commonly used uncertainty measures**
>
> The goal of our work is to create a trustworthy framework for mobile health models that achieves high performance, has reasonable computational cost, and can be integrated with existing health models without the need for retraining. In this process, we eliminated some commonly used uncertainty measures. The baseline for uncertainty measurement has been vanilla softmax probabilities, but this method is known to be susceptible to overconfidence on out-of-distribution inputs [5]. On the other hand, Bayesian approximation has also been used with methods such as Monte-Carlo Dropout or DropConnect. However, Bayesian approximation methods require a lot of computation that makes it costly at inference time [6].
>
> [1] Heart health notifications on your Apple Watch. Apple Support. (2021, April 26). https://support.apple.com/en-us/HT208931.
>
> [2] Bui, P. (2021, May 18). Using ai to help find answers to common skin conditions. Google. https://blog.google/technology/health/ai-dermatology-preview-io-2021/.
>
> [3] Patel, S. (2021, February 4). Take a pulse on health and wellness with your phone. Google. https://blog.google/technology/health/take-pulse-health-and-wellness-your-phone.
>
> [4] McHugh-Johnson, M. (2021, March 30). Sleeping on the job: How we built the new Nest hub. Google. https://blog.google/products/google-nest/sleeping-job-how-we-built-new-nest-hub/.
>
> [5] Nguyen, Anh, Jason Yosinski, and Jeff Clune. "Deep neural networks are easily fooled: High confidence predictions for unrecognizable images." Proceedings of the IEEE conference on computer vision and pattern recognition. 2015.
>
> [6]  Abdar, Moloud, et al. "A review of uncertainty quantification in deep learning: Techniques, applications and challenges." Information Fusion (2021).

---

### Decision · Program_Chairs · 2021-09-27

**Decision:**

Accept (Poster)

**Comment:**

Thanks to the authors for an interesting and thorough study of OOD methods in the healthcare context.

The reviewers all agreed that this work was well done and interesting, but the main concern is relevance for the NeurIPS community.  While I agree with reviewer Ku9s that the main interest of the this community is in ML methodology and theory, I do believe it is important to investigate real use cases of methods --- two of which were published recently in NeurIPS, and one in ICML.  MLHC, CHIL, JAMIA, and JMIR may also be appropriate venues, but the NeurIPS/ICML community developed these methods, and are an important audience for this kind of evaluation.  I tend to agree with the authors that their work falls squarely within the "Social Aspects of Machine Learning" (and "Applications"), so I am not as concerned about the relevance of this work for the ML community.

Beyond the relevance concern, all reviewers were quite positive about the work itself.  Reviewer hRAn detailed some concerns about definitions (OOD, confidence score), which should be clarified in the main text.  Reviewer QZBK also describes an improvement to the user study, which should be incorporated into the discussion.